# Phytocompounds in Precision Dermatology: COX-2 Inhibitors as a Therapeutic Target in Atopic-Prone Skin

**DOI:** 10.3390/biom15070998

**Published:** 2025-07-11

**Authors:** Muhammad Suleman, Abrar Mohammad Sayaf, Chiara Moltrasio, Paola Maura Tricarico, Francesco Giambuzzi, Erika Rimondi, Elisabetta Melloni, Paola Secchiero, Annalisa Marcuzzi, Angelo Valerio Marzano, Sergio Crovella

**Affiliations:** 1Laboratory of Animal Research Center (LARC), Qatar University, Doha P.O. Box 2713, Qatar; 2Center for Biotechnology and Microbiology, University of Swat, Swat 19200, Pakistan; 3School of Chemical Sciences, University Sains Malaysia, Gelugor 11800, Malaysia; amsayaf@gmail.com; 4Dermatology Unit, Fondazione IRCCS Ca’ Granda Ospedale Maggiore Policlinico, 20122 Milan, Italy; chiara.moltrasio@policlinico.mi.it (C.M.); angelo.marzano@unimi.it (A.V.M.); 5Department of Pediatrics, Institute for Maternal and Child Health, IRCCS “Burlo Garofolo”, 34137 Trieste, Italy; tricaricopa@gmail.com; 6Department of Advanced Diagnostics, Institute for Maternal and Child Health, IRCCS “Burlo Garofolo”, 34137 Trieste, Italy; francesco.giambuzzi@burlo.trieste.it; 7Department of Translational Medicine and LTTA Centre, University of Ferrara, 44121 Ferrara, Italy; erika.rimondi@unife.it (E.R.); elisabetta.melloni@unife.it (E.M.); paola.secchiero@unife.it (P.S.); 8Department of Translational Medicine, University of Ferrara, 44121 Ferrara, Italy; mrcnls@unife.it; 9Department of Pathophysiology and Transplantation, Università degli Studi di Milano, 20122 Milan, Italy; 10Department of Biomedical Sciences, Health Cluster, Qatar University, Doha P.O. Box 2713, Qatar

**Keywords:** atopic dermatitis, COX-2, phytocompounds, molecular dynamics simulation, pharmacokinetics, toxicity

## Abstract

Atopic dermatitis (AD) is a chronic, multifactorial inflammatory skin disease characterized by persistent pruritus, immune system dysregulation, and an increased expression of cyclooxygenase-2 (COX-2), an enzyme that plays a central role in the production of prostaglandins and the promotion of inflammatory responses. In this study, we employed a comprehensive computational pipeline to identify phytocompounds capable of inhibiting COX-2 activity, offering an alternative to traditional non-steroidal anti-inflammatory drugs. The African and Traditional Chinese Medicine natural product databases were subjected to molecular screening, which identified six top compounds, namely, Tophit1 (−16.528 kcal/mol), Tophit2 (−10.879 kcal/mol), Tophit3 (−9.760 kcal/mol), Tophit4 (−9.752 kcal/mol), Tophit5 (−8.742 kcal/mol), and Tophit6 (−8.098 kcal/mol), with stronger binding affinities to COX-2 than the control drug rofecoxib (−7.305 kcal/mol). Molecular dynamics simulations over 200 ns, combined with MM/GBSA binding free energy calculations, consistently identified Tophit1 and Tophit2 as the most stable complexes, exhibiting exceptional structural integrity and a strong binding affinity to the target protein. ADMET profiling via SwissADME and pkCSM validated the drug-likeness, oral bioavailability, and safety of the lead compounds, with no Lipinski rule violations and favorable pharmacokinetic and toxicity profiles. These findings underscore the therapeutic potential of the selected phytocompounds as novel COX-2 inhibitors for the management of atopic-prone skin and warrant further experimental validation.

## 1. Introduction

Atopic dermatitis (AD) is a chronic, immune-mediated inflammatory skin disorder, clinically characterized by relapsing eczematous eruptions and associated with chronic intractable pruritus and other IgE-related disorders, such as allergic rhinitis, asthma, and food allergies. Globally, its prevalence rates are estimated to be up to 15–25% in children and 3–7% in adults, making it one of the most prevalent dermatological diseases [1,2]. Alongside skin barrier dysfunction, AD results from a complex interplay between genetic predisposition, environmental exposure, immune dysregulation, and microbial imbalance [3,4]. In addition to these key factors, the exposome—encompassing environmental and lifestyle influences, including environmental pollutants, allergens, and stress—further shapes the immunopathogenic landscape of AD [1,5].

From a genetic perspective, filaggrin (FLG) gene mutations are among the strongest risk factors for AD, leading to compromised skin barrier integrity and increased transepidermal water loss [6]. At the same time, type 2 skin immune activation promotes a decreased expression of epidermal antimicrobial peptides and FLG, regardless of their mutational status [7]. Cytokine profiles in skin inflammation, particularly in atopic skin conditions, are skewed toward a Th2-dominant immune response, involving elevated levels of interleukin (IL)-4, IL-5, IL-13, and IL-31 and chemokines such as CC chemokine ligand (CCL) 17 and CCL22, which perpetuate inflammatory cascades and compromise skin barrier integrity through chemokine receptor signaling [8]. IL-22 might also play a role, as shown by the clinical benefit obtained with IL-22 antagonists in a subgroup of patients with high basal levels of this cytokine [9]. Furthermore, Th1/Th17 responses can also be upregulated, especially in chronic lesions, but their therapeutic potential remains to be defined [10].

Among the inflammatory mediators implicated in atopic skin conditions, cyclooxygenase-2 (COX-2) is a key inducible enzyme that catalyzes the conversion of arachidonic acid in prostaglandins, lipid mediators known to play a crucial rule in inflammation by contributing to pain, vasodilation, and leukocyte recruitment [11].

The main role of COX-2 in the pathophysiology of atopic dermatitis (AD) is related to the manifestation of chronic, intractable pruritus—a hallmark symptom that severely impairs quality of life. COX-2 expression is markedly upregulated in inflamed skin lesions in response to various inflammatory stimuli, including cytokines, allergens, and microbial antigens. This upregulation contributes not only to the amplification of the inflammatory cascade through the enhanced synthesis of pro-inflammatory prostaglandins but also to the generation of reactive oxygen species (ROS), further exacerbating local tissue damage and the sensitization of peripheral nerve endings. The resulting neuroinflammation is closely linked to the heightened perception of itch.

Moreover, COX-2-derived prostaglandins such as PGE_2_ have been shown to directly sensitize itch-related sensory neurons and interact with pruritogenic pathways, thus sustaining and intensifying the itch–scratch cycle. In this context, the pharmacological inhibition of COX-2 emerges as a compelling therapeutic strategy to mitigate both inflammation and oxidative stress. Importantly, COX-2 inhibition has the potential to break the vicious cycle of pruritus by modulating neuroimmune interactions and restoring skin barrier function. Therefore, targeting COX-2 is not only relevant for controlling inflammatory flares in atopic-prone skin but may also offer a valuable intervention to relieve the burdensome and often treatment-resistant pruritus associated with moderate-to-severe forms of AD [12,13,14].

Non-steroidal anti-inflammatory drugs (NSAIDs) and selective COX-2 inhibitors (coxibs) have shown efficacy but are limited by risks of gastrointestinal bleeding, renal toxicity, and cardiovascular events [15]. This drives the need for anti-inflammatory alternatives.

Plant-derived natural products, or phytochemicals, offer a promising alternative due to their structural diversity, biological activity, and relatively low toxicity [16]. Several medicinal plants have shown COX-2 inhibitory activity through flavonoids, alkaloids, and terpenoids [17]. The recent surge in computational methods has facilitated the virtual screening and structure-based drug discovery of natural products, accelerating early-phase candidate selection [17].

In this study, we employed a computational pipeline to identify phytocompounds with potential COX-2 inhibitory effects suitable for AD management. A phytochemical library was compiled from multiple open-access databases: the Traditional Chinese Medicine Systems Pharmacology Database (TCMSP) [18] and curated datasets of South African medicinal plants known for dermatological applications [19]. These libraries encompass thousands of well-characterized molecules with diverse ethnopharmacological backgrounds.

We apply a multi-layered virtual screening strategy involving molecular docking, pharmacokinetic and toxicity profiling (ADMET) [20], molecular dynamics (MD) simulations, and MM-PBSA binding energy calculations [21]. This study addresses the pressing clinical challenge of managing chronic inflammation and, most critically, persistent pruritus in individuals with atopic-prone skin. By prioritizing the identification of safe, potent, and selective COX-2 inhibitors derived from natural sources, our approach aims to develop novel therapeutic candidates capable of targeting both cutaneous inflammation and the underlying neuroimmune mechanisms that drive chronic itch. COX-2 plays a pivotal role not only in amplifying inflammatory cascades through prostaglandin synthesis but also in sensitizing peripheral sensory neurons via neuroinflammatory mediators, thereby contributing to the persistence and severity of pruritus. By modulating COX-2 activity, the phytocompounds identified in our study may help interrupt this neurocutaneous axis, offering a dual-action strategy to reduce both inflammatory burden and neuronal hyperexcitability. This integrative approach holds particular promise for improving clinical outcomes in patients with atopic dermatitis, especially those who are refractory to current therapies or require safer long-term treatment options

## 2. Materials and Methods

### 2.1. COX-2 3D Structure Retrieval and Minimization

The three-dimensional structure of cyclooxygenase-2 (COX-2), co-crystallized with a selective inhibitor (PDB ID: 5KIR), was retrieved from the Protein Data Bank (https://www.rcsb.org/). The structure was visualized using PyMOL software (https://pymol.org/), and the bound ligand, as well as any non-essential heteroatoms, was removed to isolate the protein. The resulting structure underwent refinement using UCSF Chimera (version 1.19) (https://www.cgl.ucsf.edu/chimera/, accessed on 30 March 2025; University of California, San Francisco, CA, USA), where hydrogen atoms were added, and energy minimization was performed to correct any geometric strain and prepare the protein for downstream docking analysis.

### 2.2. Compound Database Retrieval

To identify potential natural inhibitors of COX-2, compound libraries were sourced from five comprehensive databases: the South African Natural Product Database (SANPDB) (https://sancdb.rubi.ru.ac.za/) [22], East African Natural Products Database (EANPDB), North East African Natural Products Database (NEANPDB), North African Natural Products Database (NANPDB) (https://african-compounds.org/anpdb/, accessed on 30 March 2025) [19], and Traditional Chinese Medicine (TCM) database. These repositories offer a diverse range of natural compounds with reported medicinal properties, making them valuable resources for virtual screening in drug discovery.

### 2.3. Rule of Five Filtering

All retrieved compounds were filtered using the FAF4Drug online server (https://fafdrugs4.rpbs.univ-paris-diderot.fr/, accessed on 30 March 2025) to evaluate their drug-likeness based on Lipinski’s Rule of Five (Ro5). This filtering process considered parameters such as molecular weight, lipophilicity (log P), hydrogen bond donors and acceptors, and molecular flexibility. Compounds that violated more than one of these rules, or that contained structurally undesirable or potentially toxic groups, were excluded to retain only those with favorable pharmacokinetic characteristics [23].

### 2.4. Virtual Screening

Compounds that passed Ro5 filtering were converted into .pdbqt format using Open Babel (https://openbabel.org/index.html, accessed on 30 March 2025) to enable docking with EasyDock Vina 2.0. Virtual screening was conducted in two stages: an initial screening with an exhaustiveness value of 16 to narrow down the compound list, followed by a more rigorous secondary screening using an exhaustiveness of 64. The docking algorithm employed was based on AutoDock4, and binding affinities were calculated to rank the compounds for further analysis.

### 2.5. Induced-Fit Docking (IFD)

The top 10% of compounds from each database, as determined by virtual screening scores, were subjected to induced-fit docking (IFD) using AutoDockFR (https://ccsb.scripps.edu/adfr/, accessed on 30 March 2025). This advanced docking protocol allows for receptor flexibility and models covalent and non-covalent interactions more accurately. The use of IFD helped in identifying compounds with a strong, adaptive binding potential to the flexible COX-2 active site [24].

### 2.6. Visualization and Interaction Analysis

The six highest-ranking compounds from IFD were further analyzed using PyMOL and Schrödinger Maestro software (Release 2025‑1: Maestro, Schrödinger, LLC, New York, NY, USA). Visual inspection focused on key interactions between the ligands and COX-2 active site residues, including hydrogen bonding, hydrophobic interactions, and π-π stacking. This step provided structural insights into how each compound interacts with and potentially inhibits COX-2 activity.

### 2.7. Molecular Dynamics Simulation Analysis

Molecular dynamics (MD) simulations of drug–protein complexes were performed using Amber20 (update 12): pmemd.cuda and associated AmberTools modules; University of California, San Francisco, CA, USA [25,26]. First, the docked complexes were prepared by removing the non-essential molecules and modeling the missing amino acid residues. In parallel, the ligand structures were optimized and assigned suitable force field parameters using the antechamber module in AmberTools. Topology and coordinate files were then generated separately for both the protein and the ligand and were merged using the LEaP module to form a complete simulation system. This complex was immersed in a TIP3P water box and neutralized by adding counterions such as Na^+^ or Cl^−^. Afterward, two-stage energy minimization was carried out, starting with the steepest descent method, followed by the conjugate gradient technique, in order to relieve any steric clashes and stabilize the structure [27,28]. The system was then equilibrated by gradually increasing the temperature from 0 K to around 300 K, using a Langevin thermostat for temperature control. Pressure was regulated using barostats such as Berendsen or Andersen [29]. After successful equilibration, a 200 ns simulation was performed by using the NPT or NVT ensemble conditions [30].

### 2.8. Post-Simulation Stability, Compactness, and Residual Fluctuation Analyses

Post-simulation analyses, such as stability, compactness, and residual fluctuation analyses, were performed by processing the trajectories of RMSD, Rg, and RMSF by using the CPPTRAJ or PTRAJ modules [31,32,33,34]. The following equations were used to calculate the RMSD, Rg, and RMSF:
(1)$$RMSD=\sqrt{\frac{\sum d^{2}i=1}{N_{atoms}}}$$

The value "di" reflects how much the position of each atom shifts when comparing the original structure to its superimposed counterpart.

The radius of gyration was calculated by the following equation:
(2)$${R^{2}}_{\mathrm{gyr}}=\frac{1}{M}\sum_{i=1}^{N}m_{i}\left(r_{i}-R^{2}\right)$$
(3)$$M=\sum_{i=1}^{N}m_{i}$$
(4)$${R=N}^{-1}\sum_{i=1}^{N}r_{i}$$

The residual fluctuation was calculated by the following equation:
(5) Thermal factor or B−factor=[(8π×2)/3] (msf)

### 2.9. Binding Free Energy Analysis

Binding free energy calculations play a key role in assessing drug–protein interactions by quantifying how strongly a ligand binds to its target. Lower (more negative) values suggest a higher binding affinity, aiding in drug candidate selection, validating docking results, and optimizing leads [35,36]. Consequently, to check the binding affinity of the control and shortlisted lead compounds with COX-2, we calculated the binding free energies by using the MM/GBSA approach. Stable frames were extracted from the simulation trajectories by using the MMPBSA.py script [37]. The following equations were used to calculate the binding free energies:
(6)$${\;\;\;\;\;\;\;\;\;\;\;\;\;\;∆G}_{bind}=G_{\left(complex,\;\;\;solvated\right)}-G_{\left(ligand,\;\;\;solvated\right)}-G_{\left(COX2,\;\;\;solvated\right)}$$

However, the following equation was used to calculate the specific energy contribution:
(7)$$\;G=E_{Molecular\;Mechanics}-G_{solvated}-TS$$

To calculate the specific energy term, the formula was restructured as follows:
(8)$${∆G}_{bind}={∆E}_{Molecular\;Mechanics}+{∆G}_{solvated}-∆TS={∆G}_{vaccum}+{∆G}_{solvated}$$
(9)$${\;\;\;\;\;\;\;\;\;\;\;\;\;\;\;∆E}_{Molecular\;Mechanics}={∆E}_{int}+{∆E}_{electrostatic}+{∆E}_{vdW}$$
(10)$$\;∆G_{solvated}=∆G_{Generalized\;born}+∆G_{surface\;area}$$
(11)$$∆G_{surface\;area}=\gamma .SASA+b$$

### 2.10. Pharmacokinetic Analysis of Control and Shortlisted Compounds

Lipinski’s Rule of Five plays a fundamental role in modern drug discovery by defining essential physicochemical parameters that influence the oral bioavailability of drug candidates. According to this rule, an orally active compound generally possesses a molecular weight below 500 Da, no more than five hydrogen bond donors, no more than ten hydrogen bond acceptors, and a log P value not exceeding 5 [38]. To determine whether our selected phytocompounds conform to these drug-likeness criteria, we utilized the SwissADME platform (http://www.swissadme.ch/), an online tool that provides a comprehensive analysis of molecular properties, pharmacokinetics, and overall drug-likeness [39]. In addition to Lipinski’s assessment, we performed an extensive ADMET analysis covering absorption, distribution, metabolism, excretion, and toxicity to further evaluate the drug development potential of these compounds. Considering that pharmacokinetic shortcomings account for a significant proportion of drug development failures, such an assessment is crucial. For this purpose, we employed the pkCSM web server (https://biosig.lab.uq.edu.au/pkcsm/, accessed on 30 May 2025), which uses graph-based signatures to predict key pharmacokinetic and toxicity properties [40]. This analysis included evaluations of water solubility, gastrointestinal absorption, blood–brain barrier permeability, hepatotoxicity, and skin sensitization, providing a holistic view of each compound’s pharmacological profile.

## 3. Results and Discussion

Natural products, particularly those sourced from ethnomedicinal traditions across North and East Africa, as well as Traditional Chinese Medicine (TCM), represent a rich and diverse reservoir of biologically active compounds with therapeutic potential. In this study, we conducted a focused molecular docking investigation to explore the binding affinities and interaction profiles of six natural bioactive compounds, encompassing flavonoids, quinones, phenolics, and pyrones, against COX-2, a selected therapeutic target.

COX-2 is an inducible enzyme responsible for the biosynthesis of prostaglandins from arachidonic acid, a key step in the inflammatory signaling cascade. Upon activation by pro-inflammatory signals, COX-2 expression rises significantly, resulting in a heightened production of prostaglandins. These lipid compounds not only intensify inflammatory processes but also contribute to oxidative stress by promoting the accumulation of reactive oxygen species in affected tissues.

Rofecoxib (RCX), a clinically approved COX-2 inhibitor, was used as a reference compound to validate the docking protocol [41]. RCX demonstrated a binding affinity score of −7.305 kcal/mol, forming key hydrogen bonds with Arg513 and Hie90, two critical residues known to anchor ligands at the COX-2 active site. Figure 1 illustrates the redocking of rofecoxib (RCX) at the COX-2 active site, showing its consistent binding mode and interaction profile. The redocking results further validate the docking methodology, with a minimal RMSD of 0.073, confirming the accuracy and reliability of the simulation in reproducing the known binding interactions of RCX. Notably, all six top-ranked natural compounds identified in our study exhibited stronger binding affinities than the control, with docking scores ranging from −8.098 to −16.528 kcal/mol. These compounds were carefully selected based on their documented ethnopharmacological relevance, which includes anti-inflammatory, osteoprotective, antimicrobial, and wound healing properties. By integrating binding energy scores with detailed molecular interaction mapping, we aimed to evaluate their potential as lead compounds for drug discovery.

The following section presents a comprehensive analysis of their binding affinities, functional group interactions, and structure–activity relationships, shedding light on their pharmacological promise and guiding future optimization strategies. The top-ranked natural compounds established a variety of stabilizing interactions at the COX-2 active site, including hydrogen bonds, π–π stacking, and electrostatic interactions, primarily mediated by phenolic hydroxyl groups, carbonyl functionalities, and aromatic rings. These interactions involved key active site residues, including Arg120, Hie90, Tyr355, Ser530, Met522, Arg513, Trp387, Phe518, and Gln192, many of which are crucial for ligand anchoring, enzymatic activity, and NSAID selectivity, underscoring the potential of these compounds as strong COX-2 binders. The successful identification of Arg513 and Hie90 [42] as common anchoring residues across multiple bioactive compounds reinforces the reliability of the docking protocol and emphasizes the drug-likeness of natural scaffolds that engage in multifunctional binding patterns.

### 3.1. Virtual Screening of Phytocompounds Against COX-2

Virtual drug screening is a transformative approach in drug discovery that complements traditional methods by improving efficiency, reducing costs, and accelerating the development of new therapeutics. It plays a vital role in precision medicine, pandemic preparedness, and the development of treatments for complex diseases like cancer and infectious diseases [43]. Virtual screening identified 8-C-p-hydroxybenzylluteolin (1-NA/EA/TCM) as the first top hit, a biologically active flavonoid isolated from *Thymus hirtus*, a species of the Labiateae family, which is widely distributed in Eastern Algeria and has traditional uses in North African and East African medicine, as well as in Traditional Chinese Medicine. This compound is a polyphenolic flavonoid belonging to the glycosylated flavonol class and is known for its anti-osteoclastic activity. Pharmacological studies indicate that 8-C-p-hydroxybenzylluteolin glycosides inhibit RANKL-induced osteoclast differentiation in bone marrow-derived macrophages (BMDMs) while also downregulating osteoclast-specific genes like tartrate-resistant acid phosphatase, cathepsin K, NFATc1, and DC-STAMP [44]. These findings suggest that the compound holds significant therapeutic potential in treating bone-resorptive diseases such as osteoporosis. In a molecular docking analysis, 8-C-p-hydroxybenzylluteolin achieved the highest binding affinity score of −16.528 kcal/mol (Table 1). An interaction analysis revealed a rich interaction profile with a total of eight hydrogen bonds (HBs) and one π–π interaction with key residues at the active site of the target protein. Detailed profiling illustrated that the resorcinol OH group and dihydropyran-4-one oxygen formed hydrogen bonds with Arg120 and Tyr355, while its phenol OH and pyrocatechol moieties engaged in multiple hydrogen bonds with Met522, Hie90, Gln192, and Phe518. Furthermore, a π–π stacking interaction was observed between the resorcinol ring and Arg120, which contributed to the further stabilization of the ligand–protein complex (Figure 2). Notably, the extensive interactions with conserved Hie90 critical for COX-2 active site dynamics highlight the molecule’s potential in anti-inflammatory applications, further supported by its reported inhibition. These findings suggest 8-C-p-hydroxybenzylluteolin as a potential candidate for the development of COX-2-targeting therapeutic agents for the management of atopic-prone skin.

Furthermore, the second top hit identified in the molecular screening was Eptosphaerin D (2-NA/EA), a quinone compound isolated from *Bulbine frutescens*, a plant widely recognized not only for its ornamental appeal due to its bright yellow flowers and succulent leaves but also for its traditional medicinal use. The leaf exudate of *B. frutescens* has long been applied in the treatment of wounds and for esthetic skin care, and its gel extract has recently been patented for promoting wound healing [45]. A docking analysis revealed that Eptosphaerin D exhibited a binding affinity score of −10.879 kcal/mol, securing its position as the second most promising compound (Table 1). Despite having a relatively lower affinity than the top-ranked flavonoid, Eptosphaerin D displayed key molecular interactions with catalytically important residues of the target protein. The hydroxyl group of its methylhydroquinone moiety formed a hydrogen bond with Hie90, a conserved residue known for its role in stabilizing ligands at the COX-2 active site. Additionally, a second hydrogen bond was formed between the phenolic OH group and Ser530, while a stabilizing π–π interaction was observed between the phenol ring and Trp387 (Figure 3). These interactions suggest a stable and specific ligand–protein complex, highlighting the compound’s potential therapeutic relevance, especially in inflammation regulation and tissue repair.

Moreover, the third top hit identified in the molecular screening was 2-(p-hydroxybenzyl)-7-methoxybenzofuran-6-ol (3-NA/EA/TCM), a phenolic compound isolated from *Dorstenia kameruniana*, a species within a genus known for its ethnopharmacological relevance. Various *Dorstenia* species are traditionally used in African folk medicine to treat a wide range of ailments. For example, in Cameroon, the leaves of *Dorstenia psilurus* are used to manage cough, stomach pain, and headaches, while in northern Ethiopia, the roots of *Dorstenia barnimiana* are applied in the treatment of leprosy, liver disorders, and intestinal parasites [46]. In molecular docking studies, this compound exhibited a binding affinity score of −9.760 kcal/mol, placing it as the third most promising ligand (Table 1). Interaction profiling revealed a favorable binding orientation stabilized by a network of hydrogen bonds and π–π interactions with key active site residues. Specifically, the phenolic hydroxyl group formed a hydrogen bond with Met522, while the phenol ring participated in π–π stacking interactions with Trp387 and Phe518, enhancing complex stability. Moreover, the hydroxyl group of the 2-methoxyphenol moiety established a hydrogen bond with Arg513, and the 2-methoxyphenol ring further engaged in a π–π interaction with Tyr355 (Figure 4).

The fourth top hit identified in the molecular screening analysis was Puguenolide (4-NA/EA/TCM), a phenanthrenoid compound isolated from Uvaria puguensis, a plant species native to East Africa. Members of the Uvaria genus, particularly those found in Tanzania, are widely used in traditional medicine for the treatment of malaria and fever-related illnesses [47]. In the docking study, Puguenolide displayed a binding affinity score of −9.752 kcal/mol, closely matching the third-ranked compound and suggesting a favorable interaction with the target protein (Table 1). The interaction analysis revealed a selective binding profile characterized by two key hydrogen bonds with the conserved residue Hie90. Specifically, the carbonyl oxygen of the oxan-2-one ring and the phenolic hydroxyl group each formed hydrogen bonds with Hie90, a residue critically involved in ligand stabilization at the COX-2 active site (Figure 5). While Puguenolide demonstrated a simpler interaction profile than the top three hits, the presence of dual hydrogen bonds with such a catalytically significant residue underlines its potential biological relevance. Its traditional use in febrile and inflammatory conditions, coupled with its targeted interactions in silico, supports its promise as a candidate for further development in antipyretic or anti-inflammatory drug discovery.

The fifth top hit identified in the molecular screening analysis was 1-hydroxy-5-methoxy-3-methyl-9,10-dihydroanthracene-9,10-dione (5-NA/EA/TCM), a quinone derivative isolated from *Aloe sinkatana*, a plant species traditionally used in folk medicine for the treatment of various ailments. The leaves and their exudates are particularly valued for managing skin conditions, fever, constipation, and inflammation of the colon, as well as for their antidiabetic properties [48]. In the docking study, this compound exhibited a binding affinity score of −8.742 kcal/mol, placing it as the fifth-ranked ligand in the screening panel (Table 1). The interaction analysis revealed a triad of stabilizing contacts at the COX-2 active site. Specifically, the hydroxyl group of the methylphenol moiety formed hydrogen bonds with Arg120 and Hie90, two residues critical for ligand anchoring and enzymatic function. Additionally, the carbonyl oxygen of the cyclohexane-1,4-dione ring established a hydrogen bond with Tyr355, a residue frequently involved in ligand stabilization and NSAID binding (Figure 6). Although its binding affinity is modest relative to that of higher-ranked compounds, the engagement of essential active site residues supports its potential pharmacological activity. Together with its ethnobotanical significance, these molecular interactions suggest that 1-hydroxy-5-methoxy-3-methyl-9,10-dihydroanthracene-9,10-dione may serve as a promising lead for further exploration in AD.

The sixth top hit identified in the molecular screening analysis was Macrocarpon C (6-NA/EA/TCM), a pyrone derivative isolated from the endophytic fungus *Fusarium tricinctum*. This compound is notable for its role in inducing secondary metabolite production, a trait that has garnered interest in natural product chemistry and pharmacology [49]. In the docking study, Macrocarpon C displayed a binding affinity score of −8.098 kcal/mol, ranking sixth among the screened candidates (Table 1). Despite its comparatively lower binding energy, the compound demonstrated meaningful interactions at the active site of the target protein. Specifically, the carbonyl group of the methylpyran-4-one moiety formed a hydrogen bond with Arg513, a residue involved in substrate orientation and catalytic function. Additionally, the hydroxyl group of the resorcinol ring engaged in a stabilizing hydrogen bond with Met522, a residue known to contribute to ligand specificity and stabilization (Figure 7). Although the interaction profile is less extensive than that of higher-ranked compounds, the selective engagement with catalytically significant residues suggests potential for biological activity. Combined with its unique fungal origin and its influence on secondary metabolite biosynthesis, Macrocarpon C holds promise as a molecular scaffold for future optimization in AD drug discovery.

All in all, the docking study presented herein underscores the pharmacological potential of traditionally used natural compounds, with multiple candidates demonstrating compelling binding interactions with key catalytic residues of the target protein. Of particular interest, 8-C-p-hydroxybenzylluteolin not only exhibited the most favorable docking score (–16.528 kcal/mol) but also formed multiple hydrogen bonds and π–π interactions, prominently engaging Arg120, Tyr355, Met522, and, crucially, Hie90 and Gln192, suggesting high affinity and stable binding at the active site. Similarly, Eptosphaerin D and 2-(p-hydroxybenzyl)-7-methoxybenzofuran-6-ol displayed specific interactions with Hie90 and Arg513, mirroring the binding orientation of rofecoxib, thus validating their potential as COX-2 modulators or anti-inflammatory leads. The ability of these natural ligands to occupy the binding pocket and interact with conserved residues like Arg513 and Hie90, known to be central in COX-2 inhibition, highlights their therapeutic relevance. Although rofecoxib exhibited a lower binding affinity (–7.305 kcal/mol), its selectivity is well-documented, and the fact that several natural ligands outperformed it energetically while targeting the same residues is both surprising and promising. To further validate the binding stability of the identified compounds, we processed the top three hits for a molecular dynamics simulation analysis.

### 3.2. Molecular Dynamics Stability Analysis of Tophit-COX-2 Complexes

The calculation of post-simulation Root Mean Square Deviation (RMSD) is crucial in drug–protein interaction studies, as it provides a quantitative measure of the structural stability and conformational changes in the protein–ligand complex during the molecular dynamics simulation. By evaluating the RMSD over time, researchers can assess whether the complex remains stable or undergoes significant deviations, which may indicate weak or unstable binding [50,51]. The RMSD (Root Mean Square Deviation) analysis presented in Figure 8 provides detailed insights into the conformational stability of the COX-2 protein when complexed with the control ligand and the three top hit compounds identified through virtual screening over a 200-nanosecond molecular dynamics (MD) simulation. In the case of the control-COX-2 complex, the RMSD begins with an initial rise as the system equilibrates and then stabilizes around 2.0–2.5 Å up to approximately 150 ns, indicating a relatively stable protein–ligand interaction. However, a significant increase in the RMSD is observed after 150 ns, peaking near 5.5 Å, which suggests a loss of structural integrity or conformational rearrangement of the protein possibly due to a weak binding affinity or dissociation of the control ligand, leading to increased protein flexibility. This late-stage instability may reflect an insufficient capacity of the control compound to maintain stable interactions with COX-2 (Figure 8a). However, in the case of the Tophit1-COX-2 complex, the RMSD increases gradually and fluctuates mildly between 1.0 and 2.5 Å until 125 ns; however, after this point, the RMSD drops to 1.5 Å, with a relatively stable trajectory until the end of the simulation. These moderate fluctuations are within the acceptable range for a stably bound complex and likely arise from minor side-chain or loop movements, suggesting that Tophit1 forms relatively strong interactions with COX-2 and stabilizes its structure over time (Figure 8b). Furthermore, the Tophit2-COX-2 complex exhibits the most stable RMSD profile, with values ranging between 1.5 and 2.2 Å for the entire duration of the simulation. This minimal deviation indicates a robust and consistent binding mode of Tophit2 to COX-2, leading to very limited structural perturbation. The absence of any significant drift or fluctuation implies that Tophit2 fits well into the binding pocket, possibly forming stable hydrogen bonds and hydrophobic interactions that anchor the ligand and restrict excessive movement within the protein structure (Figure 8c). In contrast, the Tophit3-COX-2 complex shows a gradual increase in the RMSD after ~120 ns, ultimately reaching close to 4.0 Å (Figure 8d). This rise suggests the occurrence of slow but notable conformational changes in the protein, which may be due to weaker binding interactions or partial disengagement of Tophit3 from the binding site. These late-stage deviations imply that, while Tophit3 can initially stabilize the protein, its long-term interaction may be less favorable than that of Tophit1 and Tophit2. Overall, the analysis highlights that Tophit2 imparts the greatest structural stability to the COX-2 protein, likely due to strong and persistent binding interactions.

### 3.3. Dynamic Residual Fluctuation Analysis of Tophit-COX-2 Complexes

The calculation of the Root Mean Square Fluctuation (RMSF) is important in drug–protein interaction studies, as it provides insights into the flexibility and mobility of individual amino acid residues within the protein during molecular dynamics simulations. By identifying regions with high RMSF values, researchers can detect flexible or disordered segments, which may play roles in ligand binding or allosteric regulation [52,53]. The Root Mean Square Fluctuation (RMSF) analysis illustrates the flexibility of each residue in the COX-2 protein across four different molecular dynamics simulations: control-COX-2 (blue) and COX-2 complexed with the top three virtual screening hits Tophit1 (red), Tophit2 (olive green), and Tophit3 (purple). The control complex exhibits notably higher RMSF values, particularly within the N-terminal region (residues ~1–100), indicating increased flexibility or instability. In contrast, all three top hit ligands significantly reduce fluctuations in this region, suggesting a stabilizing effect on the COX-2 structure upon binding. Among the drug-bound complexes, Tophit2 demonstrates the lowest overall fluctuations, indicating a more rigid and stable protein conformation, which may imply a stronger or more specific binding interaction. Moderate stabilization is observed with Tophit1 and Tophit3, which also reduce fluctuations compared to the control but to a slightly lesser extent than Tophit2. Beyond residue 100, all systems show relatively low and comparable RMSF values, suggesting that ligand binding primarily impacts the flexibility of the N-terminal and possibly the active or binding sites (Figure 9). Overall, this RMSF analysis supports the notion that the shortlisted compounds, especially Tophit2, enhance the structural stability of COX-2, potentially correlating with effective binding and inhibitory potential.

### 3.4. Post-Simulation Compactness Analysis of Tophit-COX-2 Complexes

The radius of gyration (Rg) indicates the compactness of a protein structure during a simulation; stable Rg values suggest structural integrity, while fluctuations may reflect conformational changes due to drug binding [54,55]. The radius of gyration (Rg) analysis provides insights into the overall compactness and structural stability of the COX-2 protein with the control and top three virtual screening hits. In the control-COX-2 complex, Rg fluctuates within a broader range (approximately 24.0–25.0 Å), indicating a relatively dynamic and less compact structure over the 200 ns simulation (Figure 10a). In contrast, the Tophit1-COX-2 complex shows moderate fluctuations around a narrower Rg range (~23.8–24.4 Å), suggesting slightly improved compactness and stability compared to the control (Figure 10b). The Tophit2-COX-2 system exhibits the most consistent and stable Rg values, remaining tightly clustered around ~24.0–24.3 Å throughout the simulation, implying that Tophit2 binding maintains a highly compact and stable protein conformation (Figure 10c). Similarly, the Tophit3-COX-2 complex maintains a relatively stable Rg profile but with slightly more fluctuations than Tophit2, particularly after ~150 ns (Figure 10d). Overall, all ligand-bound systems enhance the compactness of COX-2 relative to the control, with Tophit2 again emerging as the most promising candidate by promoting the highest degree of structural integrity during the simulation.

### 3.5. Post-Simulation Hydrogen Bond Analysis

A post-simulation hydrogen bond analysis is crucial in drug–protein interaction studies because it helps evaluate the stability, specificity, and strength of the binding between the ligand and the target protein [56,57]. The post-simulation hydrogen bond (H-bond) analysis presented in Figure 11 illustrates the comparative behavior of the H-bond networks in the COX-2 complexes, namely, the control versus the three top virtual hits, during extended (200 ns) MD simulations. The control-COX-2, Tophit1, and Tophit3 systems show similar numbers of hydrogen bonds, starting at around 290 and then gradually decreasing until the end of the simulation time period. However, in the case of Tophit2, the system maintains a relatively stable H-bond count, with no fluctuations throughout the simulation time period. The average numbers of hydrogen bonds for the control, Tophit1, Tophit2, and Tophit3 are recorded to be 259, 264, 264, and 265, respectively (Figure 11a–d). Overall, while all three top hits interact with the protein, Tophit2 appears to preserve the native H-bond network most effectively, indicating better structural compatibility with COX-2, potentially making it a more promising candidate for further investigation.

### 3.6. Binding Free Energy Calculations of Control and Top Hit Compound-COX-2 Complexes

The calculation of binding free energies is vital in drug–protein interaction studies, as it quantitatively estimates the strength and stability of the interaction between a ligand and its target protein. A more negative binding free energy indicates a stronger and more favorable binding affinity, which is crucial for drug efficacy. This calculation helps prioritize potential drug candidates, validate docking results, and guide lead optimization [35,36]. Therefore, to analyze the binding strength of the lead compound-COX-2 complexes, we calculate the binding free energies by using the MM/GBSA approach. The MM/GBSA-based binding free energy analysis of the control and shortlisted drug-COX-2 complexes offers valuable insights into their interaction strengths and stability. Among the shortlisted compounds, the Tophit1-COX-2 complex demonstrates the most favorable binding, with a ΔG total of −50.312 kcal/mol, which is significantly lower (more negative) than that of Tophit2 (−36.5153 kcal/mol), Tophit3 (−42.3606 kcal/mol), and especially the control-COX-2 complex (−3.0885 kcal/mol). This substantial difference indicates that Tophit1 forms the most stable and energetically favorable complex with COX-2, suggesting its superior potential as a COX-2 inhibitor. Breaking down the energy components, Tophit1 exhibits a strong van der Waals interaction (ΔEvdw = −54.4187 kcal/mol), which indicates effective hydrophobic and dispersion interactions between Tophit1 and the COX-2 active site residues. Its electrostatic energy (ΔEele = −185.7878 kcal/mol) is markedly more favorable than that of Tophit2 (−30.518 kcal/mol), Tophit3 (−4.1429 kcal/mol), and the control (−9.7443 kcal/mol), indicating the presence of strong charge–charge or polar interactions stabilizing the Tophit1-COX-2 complex. However, the solvation energy (EGB = 195.9706 kcal/mol) of Tophit1 is notably higher than that of Tophit2 (−10.8805 kcal/mol) and Tophit3 (10.6177 kcal/mol), implying that Tophit1 faces greater desolvation penalties upon binding. However, these unfavorable solvation effects are effectively compensated by the highly favorable gas-phase interaction energy (Delta G Gas = −240.2065 kcal/mol), leading to an overall strong binding affinity. The ESURF term (−6.0761 kcal/mol) further contributes to Tophit1’s stability by representing favorable nonpolar solvation contributions (e.g., cavity formation and surface tension effects). Compared to the control and other shortlisted compounds, Tophit1 shows a better balance between gas-phase interactions and solvation penalties, resulting in its dominant binding free energy profile. In stark contrast, the control-COX-2 complex shows the weakest binding (ΔG total = −3.0885 kcal/mol), with less favorable van der Waals (ΔEvdw = −44.7286 kcal/mol) and electrostatic interactions (ΔEele = −9.7443 kcal/mol), along with a substantial solvation energy penalty (EGB = 57.3348 kcal/mol). This suggests poor binding stability and interaction strength, reaffirming the superior binding performance of the shortlisted compounds, especially that of Tophit1. Overall, this detailed binding free energy profile strongly indicates that shortlisted compounds have the highest binding affinity and most stable interaction with COX-2, making it a promising lead candidate for further optimization and development as a potential COX-2 inhibitor in the management of atopic-prone skin. A list of the binding free energies is shown in Table 2.

### 3.7. Pharmacokinetic Properties Analysis of Control and Lead Compounds

A pharmacokinetic properties analysis is essential in drug development, as it evaluates how a compound is absorbed, distributed, metabolized, and excreted (ADMET) in the body. This analysis helps predict a drug’s bioavailability, half-life, potential toxicity, and ability to reach its target site at therapeutic concentrations [58,59]. Both the control and shortlisted lead compounds are evaluated using Lipinski's Rule of Five, which predicts the likelihood of good oral bioavailability. All compounds meet the criteria, showing no violations, indicating favorable drug-like properties. Specifically, the molecular weights of all compounds range from 270.284 to 391.355 Da, well below the 500 Da threshold, suggesting efficient permeability and absorption. The number of hydrogen bond acceptors varies from 4 to 7, and that of hydrogen bond donors ranges from 1 to 4, both within the acceptable limits of 10 and 5, respectively, ensuring balanced solubility and membrane permeability. Additionally, the consensus log P values, which indicate lipophilicity, are between 2.23 and 3.72, well under the cutoff of 5, suggesting a good balance between hydrophilicity and lipophilicity, essential for oral bioavailability. All compounds, including the control, meet Lipinski’s criteria with zero violations, suggesting comparable drug-likeness and oral bioavailability potential. However, the leads, especially Tophit2 and Tophit3, exhibit a lower molecular weight and a higher hydrogen bonding capacity, which may influence their solubility and biological interactions differently from the control, possibly offering improved pharmacokinetic or binding properties (Table 3).

Furthermore, the pharmacokinetic properties such as the absorption, distribution, metabolism, excretion, and toxicity of the control and shortlisted compounds are analyzed. In terms of absorption, all compounds exhibit acceptable water solubility, with Tophit2 (−3.237) being the most soluble and the control (−4.663) being the least. Caco-2 permeability is positive for the control (0.847), Tophit2 (0.84), and Tophit3 (1.053), while Tophit1 shows a negative value (−0.135), suggesting poor membrane permeability. Despite this, intestinal absorption is high across all compounds (>83%), with Tophit2 showing the highest absorption (95.145%). Regarding distribution, Tophit3 displays the highest volume of distribution (VDss = 0.318), suggesting better tissue distribution than the control (−0.372). Only Tophit3 shows blood–brain barrier (BBB) permeability, indicating potential CNS activity, although all compounds are predicted to have low CNS permeability. Regarding metabolism, only the control is a substrate for CYP3A4, while all compounds inhibit CYP1A2 and CYP2C19; however, Tophit2 does not inhibit CYP3A4, unlike the others. Regarding excretion, the control has the highest total clearance (0.768), while Tophit2 shows the lowest (0.061), indicating slower elimination. None are substrates for renal OCT2, suggesting limited renal excretion involvement. Toxicity profiles reveal that Tophit1 and Tophit2 are AMES toxic, with Tophit1 also causing skin sensitization, whereas the control and Tophit3 are non-toxic in these aspects. All compounds are non-hepatotoxic, non-carcinogenic, and mostly safe regarding respiratory toxicity, except for Tophit1, which is predicted to be toxic. Tophit3 exhibits the highest maximum tolerated dose (0.55 log mg/kg/day) among all compounds, indicating superior tolerability compared to the control (0.201 log mg/kg/day). Overall, the Tophits demonstrate a favorable balance of absorption, distribution, safety, and metabolic stability compared to the control (Table 4).

## 4. Conclusions

This study employed a rigorous and integrative computational pipeline to identify novel plant-derived inhibitors targeting cyclooxygenase-2 (COX-2), a key enzyme that orchestrates inflammatory responses and contributes to disease chronicity in atopic-prone skin. While numerous biological therapies—such as dupilumab, tralokinumab, and lebrikizumab—have transformed the therapeutic landscape of moderate-to-severe atopic dermatitis (AD), particularly by targeting type 2 inflammation, a significant proportion of patients continue to experience distressing symptoms, such as unremitting pruritus. This persistent itch, often resistant to conventional treatments, remains a major unmet clinical need and substantially impacts patient quality of life.

In this context, our study specifically addressed the challenge of chronic itch by focusing on COX-2 inhibition as a complementary anti-inflammatory and anti-pruritic strategy. Through the virtual screening of natural product libraries, followed by molecular docking, molecular dynamics simulations, MM/GBSA binding free energy calculations, and comprehensive ADMET profiling, we identified six phytochemicals with potent COX-2 inhibitory potential, excellent structural stability, and favorable drug-like properties—demonstrating superior performance compared to the clinically used COX-2 inhibitor rofecoxib.

Among these, 8-C-p-hydroxybenzylluteolin and Eptosphaerin D stood out as lead candidates due to their strong binding affinities, consistent interaction with catalytically essential residues (Arg513 and His90), and robust pharmacokinetic profiles. These interactions provide a mechanistic basis for their isoform-selective inhibition of COX-2 and suggest potential for the targeted modulation of cutaneous inflammation in AD.

The significance of these findings is underscored in the context of atopic-prone skin, where chronic inflammation, barrier disruption, oxidative stress, and neuroimmune dysfunction perpetuate the itch–scratch cycle. Elevated COX-2 activity is known to exacerbate this cycle through the production of prostaglandins and reactive oxygen species, contributing not only to skin inflammation but also to peripheral nerve sensitization. As such, the identification of phytocompounds capable of selectively inhibiting COX-2 while avoiding the adverse cardiovascular, gastrointestinal, and renal side effects of synthetic inhibitors offers a promising path toward safer, long-term interventions for individuals with AD—especially those with comorbidities or contraindications to biologic agents.

Moreover, the therapeutic relevance of our findings may extend beyond dermatology. Given the central role of COX-2 in numerous chronic inflammatory and neoplastic diseases, including rheumatoid arthritis, inflammatory bowel disease, and various cancers, the lead compounds identified here may serve as versatile scaffolds for developing multi-target or multi-disease agents. The computational strategy presented is highly adaptable and can be leveraged to accelerate drug discovery and repurposing efforts in other COX-2-mediated disorders, aligning with current priorities in precision medicine and natural compound-based therapeutics.

## Figures and Tables

**Figure 1 biomolecules-15-00998-f001:**
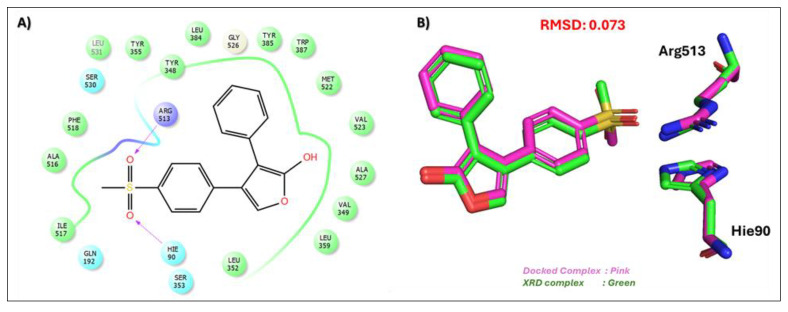
Validation of the docking protocol by the redocking of rofecoxib (RCX) at the COX-2 active site. (**A**) A 2D interaction profile of rofecoxib (RCX) at the COX-2 active site, highlighting key interactions with Arg513 and Hie90. (**B**) A 3D superimposed structure of the redocked rofecoxib (RCX), with an RMSD value of 0.073, confirming the accuracy and consistency of the docking protocol.

**Figure 2 biomolecules-15-00998-f002:**
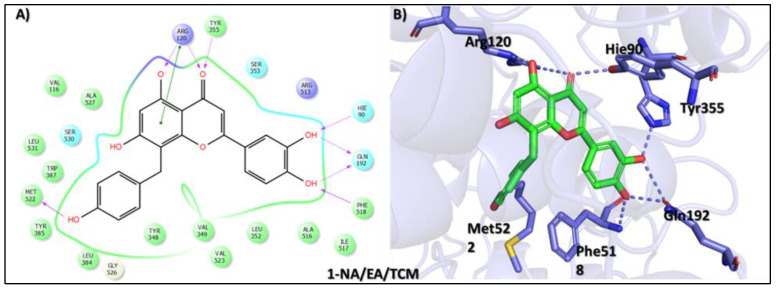
Docking mode of 8-C-p-hydroxybenzylluteolin (1-NA/EA/TCM) in the active pocket of COX-2. (**A**) Two-dimensional interaction profile, showing key interactions with residues. (**B**) Three-dimensional interaction visualization, highlighting the ligand’s binding pose and critical interactions at the COX-2 active site.

**Figure 3 biomolecules-15-00998-f003:**
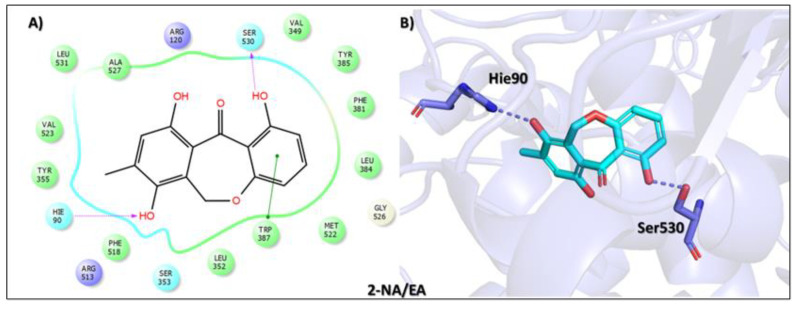
Docking mode of Eptosphaerin D (2-NA/EA) in the active pocket of COX-2. (**A**) Two-dimensional interaction profile, showing key interactions with residues. (**B**) Three-dimensional interaction visualization, highlighting the ligand’s binding pose and critical interactions at the COX-2 active site.

**Figure 4 biomolecules-15-00998-f004:**
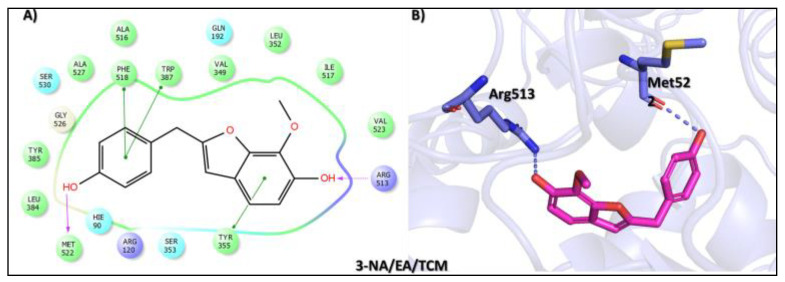
Docking mode of 2-(p-hydroxybenzyl)-7-methoxybenzofuran-6-ol (3-NA/EA/TCM) in the active pocket of COX-2. (**A**) Two-dimensional interaction profile, showing key interactions with residues. (**B**) Three-dimensional interaction visualization, highlighting the ligand’s binding pose and critical interactions at the COX-2 active site.

**Figure 5 biomolecules-15-00998-f005:**
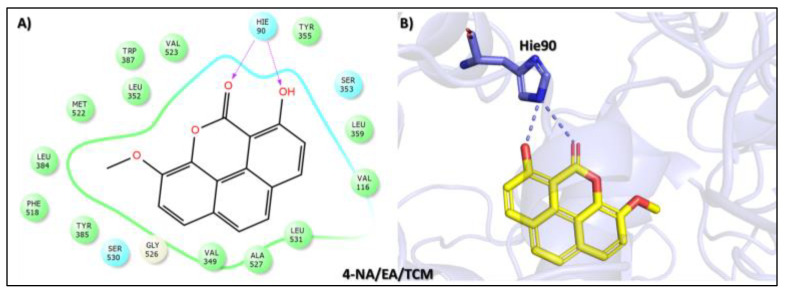
Docking mode of Puguenolide (4-NA/EA/TCM) in the active pocket of COX-2. (**A**) Two-dimensional interaction profile, showing key interactions with residues. (**B**) Three-dimensional interaction visualization, highlighting the ligand’s binding pose and critical interactions at the COX-2 active site.

**Figure 6 biomolecules-15-00998-f006:**
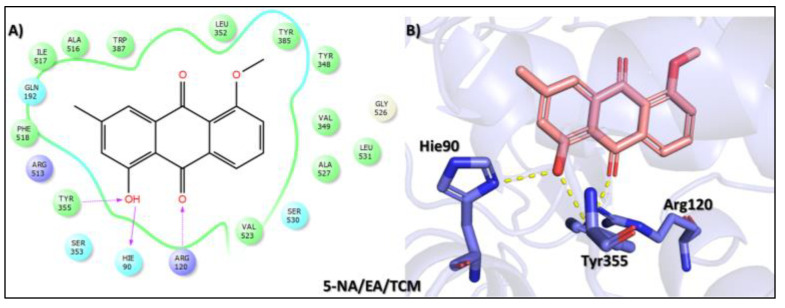
Docking mode of 1-hydroxy-5-methoxy-3-methyl-9,10-dihydroanthracene-9,10-dione (5-NA/EA/TCM) in the active pocket of COX-2. (**A**) Two-dimensional interaction profile, showing key interactions with residues. (**B**) Three-dimensional interaction visualization, highlighting the ligand’s binding pose and critical interactions at the COX-2 active site.

**Figure 7 biomolecules-15-00998-f007:**
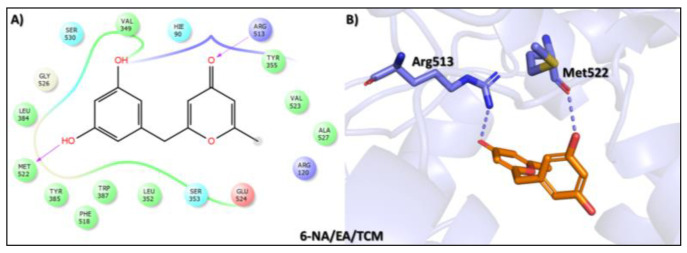
Docking mode of Macrocarpon C (6-NA/EA/TCM) in the active pocket of COX-2. (**A**) Two-dimensional interaction profile, showing key interactions with residues. (**B**) Three-dimensional interaction visualization, highlighting the ligand’s binding pose and critical interactions at the COX-2 active site.

**Figure 8 biomolecules-15-00998-f008:**
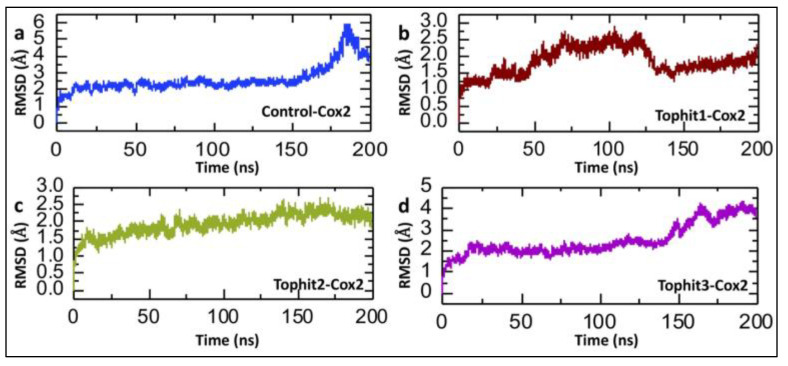
Dynamic stability analysis of control and shortlisted drug-COX-2 complexes. (**a**) The dynamic stability of the control-COX-2 complex during the 200 ns simulation. (**b**) The dynamic stability of the Tophit1-COX-2 complex during the 200 ns simulation. **(c**) The dynamic stability of the Tophit2-COX-2 complex during the 200 ns simulation. (**d**) The dynamic stability of the Tophit3-COX-2 complex during the 200 ns simulation.

**Figure 9 biomolecules-15-00998-f009:**
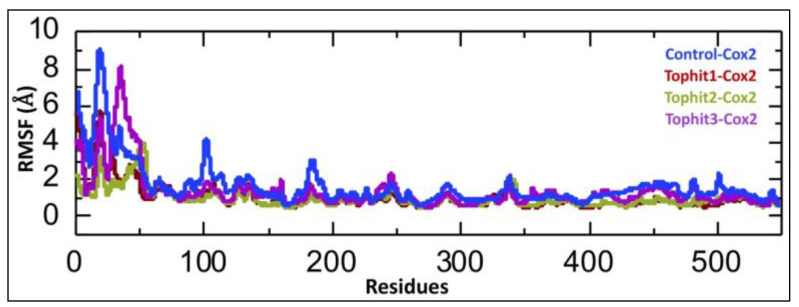
Residual fluctuation analysis of control and shortlisted drug-COX-2 complexes. Blue line represents the fluctuation of individual residues in the control-COX-2 complex during the 200 ns simulation. Red line represents the fluctuation of individual residues in the Tophit1-COX-2 complex during the 200 ns simulation. Green line represents the fluctuation of individual residues in the Tophit2-COX-2 complex during the 200 ns simulation. Purple line represents the fluctuation of individual residues in the Tophit3-COX-2 complex during the 200 ns simulation.

**Figure 10 biomolecules-15-00998-f010:**
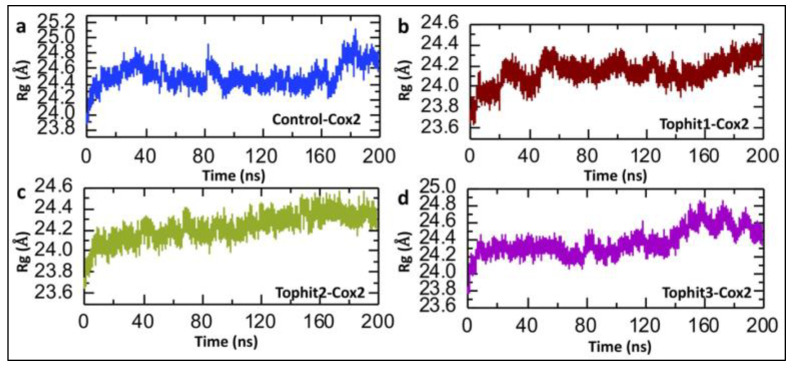
Compactness analysis of control and shortlisted drug-COX-2 complexes. (**a**) The compactness of the control-COX-2 complex during the 200 ns simulation. (**b**) The compactness of the Tophit1-COX-2 complex during the 200 ns simulation. (**c**) The compactness of the Tophit2-COX-2 complex during the 200 ns simulation. (**d**) The compactness of the Tophit3-COX-2 complex during the 200 ns simulation.

**Figure 11 biomolecules-15-00998-f011:**
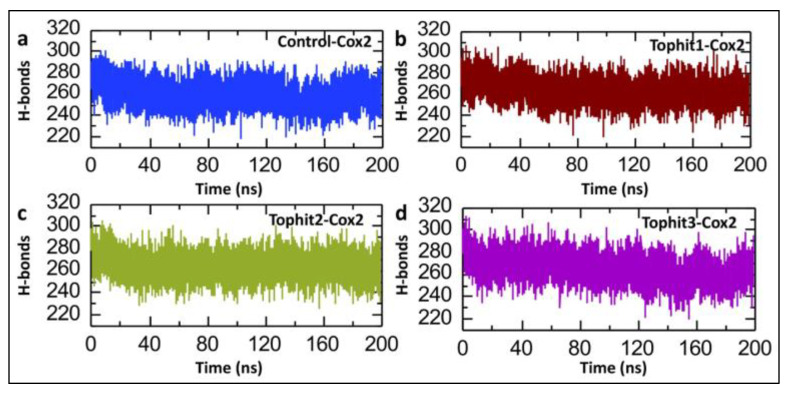
Post-simulation average hydrogen bond analysis of control and shortlisted drug-COX-2 complexes. (**a**) Hydrogen bond dynamics between the control compound and COX-2, tracked across the 200-nanosecond simulation. (**b**) The trajectories of the hydrogen bonds of the Tophit1-COX-2 complex during the 200 ns simulation. (**c**) The trajectories of the hydrogen bonds of the Tophit2-COX-2 complex during the 200 ns simulation. (**d**) The trajectories of the hydrogen bonds of the Tophit3-COX-2 complex during the 200 ns simulation.

**Table 1 biomolecules-15-00998-t001:** List of the top hit compounds with their databases, IDs, 2D structures, scientific names, docking scores, interacting residues, bond distances, and interaction types (hydrogen bonding: HB; pi–pi interaction: π-π).

Compound Name, Code, and Structure	Docking Score	Interacting Atom/FG	Interacting Residues	Interaction Nature
Control (Rofecoxib (RCX)) [42] 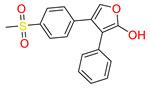	−7.305	O(Methylsulfonyl)	Arg513	HB
		O(Methylsulfonyl)	Hie90	HB
Tophit 1 (8-C-p-hydroxybenzylluteolin) [44] 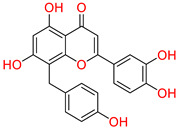	−16.528	OH(Resorcinol)	Arg120	HB
		Resorcinol ring	Arg120	Pi-Pi
		O(Dihydropyran-4-one)	Arg120	HB
		O(Dihydropyran-4-one)	Try355	HB
		OH(Phenol)	Met522	HB
		OH(Pyrocatechol)	Hie90	HB
		OH(Pyrocatechol)	Gln192	HB
		OH(Pyrocatechol)	Gln192	HB
		OH(Pyrocatechol)	Phe518	HB
Tophit 2 (Eptosphaerin D) [45] 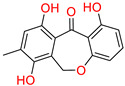	−10.879	OH(Methylhydroquinone)	Hie90	HB
		OH(Phenol)	Ser530	HB
		Phenol ring	Trp387	Pi-Pi

Tophit 3 (2-(p-hydroxybenzyl)-7-methoxybenzofuran-6-ol) [46] 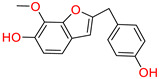	−9.760	OH(Phenol)	Met522	HB
		Phenol ring	Trp387	Pi-Pi
		Phenol ring	Phe518	Pi-Pi
		OH(2-methoxyphenol)	Arg513	HB
		2-methoxyphenol	Tyr355	Pi-Pi
Tophit 4 (Puguenolide) [47] 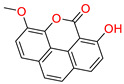	−9.752	O(Oxan-2-one)	Hie90	HB
		OH(Phenol)	Hie90	HB
Tophit 5 (1-hydroxy-5-methoxy-3-methyl-9,10 dihydroanthracene 9,10-dione) [48] 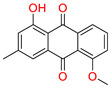	−8.742	OH(Methylphenol)	Arg120	
		OH(Methylphenol)	Hie90	
		CO(Cyclohexane-1,4-dione)	Tyr355	
Tophit 6 (Macrocarpon C) [49] 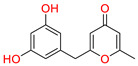	−8.098	CO(Methylpyran-4-one)	Arg513	HB
		OH(Resorcinol)	Met522	HB

**Table 2 biomolecules-15-00998-t002:** List of binding free energies of the control and shortlisted drug-COX-2 complexes calculated using the MM/GBSA approach.

MM/GBSA				
Parameters	Tophit1-COX-2	Tophit2-COX-2	Tophit3-COX-2	Control-COX-2
ΔEvdw	−54.4187 ± 0.31	2.4877 ± 0.12	−44.3097 ± 0.02	−44.7286 ± 0.23
ΔEele	−185.7878 ± 1.20	−30.518 ± 0.03	−4.1429 ± 0.13	−9.7443 ± 0.29
EGB	195.9706 ± 1.10	−10.8805 ± 0.02	10.6177 ± 0.12	57.3348 ± 0.28
ESURF	−6.0761 ± 0.01	2.3955 ± 0.00	−4.5258 ± 0.01	−5.9503 ± 0.01
Delta G Gas	−240.2065 ± 1.25	−28.0303 ± 0.14	−48.4525 ± 0.21	−54.473 ± 0.32
Delta G Solv	189.8945 ± 1.10	−8.485 ± 0.02	6.0919 ± 0.12	51.3845 ± 0.28
∆G Total	−50.312 xB1; 0.34	−36.5153 ± 0.14	−42.3606 ± 0.20	−3.0885 ± 0.32

**Table 3 biomolecules-15-00998-t003:** Lipinski's Rule of Five evaluation of the control and lead phytocompounds.

Drugs ID	Molecular Weight	Hydrogen Acceptors	Hydrogen Donors	Consensus Log P	Lipinski’s Rule	
					Results	Violation
Control-COX-2	314.362	4	1	3.72	Yes	0
Tophit1-COX-2	391.355	7	4	2.94	Yes	0
Tophit2-COX-2	272.256	5	3	2.23	Yes	0
Tophit3-COX-2	270.284	4	2	3.4434	Yes	0

**Table 4 biomolecules-15-00998-t004:** List of control and lead compounds with ADMET properties.

Properties	Control-COX-2	Tophit1-COX-2	Tophit2-COX-2	Tophit3-COX-2
**Absorption**				
Water solubility log S	−4.663	−3.605	−3.237	−3.246
Caco-2 permeability × 10^−6^	0.847	−0.135	0.84	1.053
Human intestinal absorption (%)	93.2	83.276	95.145	91.873
**Distribution**				
VDss (human)	−0.372	−1.276	0.13	0.318
BBB permeability	No	No	No	Yes
CNS permeability	−1.881	−2.398	−2.209	−2.052
Subcellular localization	Mitochondria	Mitochondria	Mitochondria	Mitochondria
**Metabolism**				
CYP2D6 substrate	No	No	No	No
CYP3A4 substrate	Yes	No	No	No
CYP1A2 inhibitor	Yes	Yes	Yes	Yes
CYP2C19 inhibitor	Yes	Yes	Yes	Yes
CYP3A4 inhibitor	Yes	Yes	No	Yes
**Excretion**				
Total clearance	0.768	0.353	0.061	0.359
Renal OCT2 substrate	No	No	No	No
**Toxicity**				
AMES toxicity	No	Yes	Yes	No
Skin sensitization	No	Yes	No	No
Hepatotoxicity	No	No	No	No
Carcinogenic	No	No	No	No
Respiratory diseases	Safe	Toxic	Safe	Safe
Max. tolerated dose (log mg/kg/day)	0.201	0.428	0.539	0.55

## Data Availability

The original contributions presented in this study are included in the article. Further inquiries can be directed to the corresponding authors.

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
