# Peer review of "Phytocompounds in Precision Dermatology: COX-2 Inhibitors as a Therapeutic Target in Atopic-Prone Skin"

_biomolecules, 2025, doi:10.3390/biom15070998_

Round 1
Reviewer 1 Report
Comments and Suggestions for Authors
The authors employed a comprehensive computational pipeline to identify phytocompounds capable of inhibiting COX-2 activity, offering an alternative to traditional non-steroidal anti-inflammatory drugs. There are some issues regarding to the submitted manuscript:
1- The authors should declare the procedures of extraction.
2- The biological work is missed.
3- SAR Structure Activity Relationship should be included in R&D section.
4- Introduction section should include the compression between the anti-inflammatory commercial drug and the new extracted ones.
5- There are some grammatical mistakes.
6- References should be updated.
7- Some references should be added to the manuscript
https://doi.org/10.3390/ph18030335
10.6026/97320630016753 https://doi.org/10.1186/s13065-023-00924-3 https://doi.org/10.1016/j.chphi.2024.1005098- Some pharmacokinetic properties are missed such as TPSA and rotatble bonds.
Author Response
Reviewer 1
The English could be improved to more clearly express the research.
Response: We improved the English language accordingly.
1- The authors should declare the procedures of extraction.
Response: We sincerely thank the reviewer for their comment regarding the extraction procedures. We would like to respectfully clarify that this study is entirely computational, as explicitly stated in the title, abstract, and throughout the manuscript. The objective of our work was to identify novel plant-derived inhibitors of COX-2 using in silico methodologies, including molecular docking, ADMET profiling, and molecular dynamics simulations.
As such, no physical extraction or wet-lab experimental work was performed, and including such procedures would fall outside the defined scope of this computational investigation. Through our manuscript it is clear that this is a purely in silico screening study, and all compound data were obtained from publicly available databases.
2- The biological work is missed.
Response: We thank the reviewer for this observation. As stated in the title and throughout the manuscript, the current study is focused on a computational drug discovery approach aimed at identifying potential COX-2 inhibitors from plant-derived compounds. Our work leverages virtual screening, molecular docking, ADMET profiling, and molecular dynamics simulations to prioritize candidates for further investigation.
We fully acknowledge that biological (in vitro or in vivo) validation was not included in this manuscript. However, this is consistent with the exploratory and predictive nature of in silico research. The compounds identified and characterized here are intended to serve as leads for future experimental studies, which we are currently planning as a follow-up to this computational phase.
3- SAR Structure Activity Relationship should be included in R&D section.
As requested, we performed a Structure–Activity Relationship (SAR) analysis, and the corresponding data are presented in our response to the reviewer. However, we chose not to include this analysis in the main manuscript, as we believe it does not significantly enhance the value of our purely computational study. SAR analysis is typically most informative when guided by experimental (bench) work, serving to interpret or predict biological outcomes. In the absence of such experimental validation, we felt that including the SAR data in the manuscript might be premature and potentially misleading. Nonetheless, we are grateful for the suggestion and have ensured the analysis is available in the review process for transparency.
Structure activity relationship (SAR) analysis
The docking analysis revealed clear structure–activity relationships (SAR) among the evaluated compounds, highlighting the influence of specific functional groups and molecular frameworks on their binding affinity toward the COX-2 active site. The structural contributions to COX-2 binding affinity are summarized in Figure 1, showcasing the critical functional groups and scaffolds identified through SAR analysis. Similarly, a detailed overview of the critical structural motifs and their corresponding influence on COX-2 binding interactions is presented in Table 1, summarizing the SAR insights derived from the docking results. The reference compound, Rofecoxib (RCX), exhibited a moderate docking score of –7.305 kcal/mol, forming hydrogen bonds with Arg513 and Hie90 through its methylsulfonyl group. Although these interactions are typical of COX-2 selectivity, the molecule lacked diverse binding interactions, limiting its affinity compared to the more structurally complex natural analogs.
Figure 1. Structure–Activity Relationship (SAR) map showing key features influencing COX-2 binding among tested natural compounds and Rofecoxib. Polyhydroxyphenyl groups, pyran one, di-one phenol, catechol and resorcinol motifs contribute to enhanced interactions with active site residues.
Among the natural compounds, 8-C-p-hydroxybenzylluteolin (1-NA/EA/TCM) demonstrated the most potent binding with a docking score of –16.528 kcal/mol. This remarkable affinity is attributed to its densely functionalized flavonoid scaffold, which enables extensive hydrogen bonding through multiple hydroxyl groups on resorcinol, pyrocatechol, and phenol moieties. Notably, the compound interacted with critical residues such as Arg120, Tyr355, Met522, Hie90, Gln192, and Phe518. In addition to polar interactions, π–π stacking with Arg120 and hydrophobic contacts further enhanced its binding strength. The presence of a dihydropyran-4-one moiety also contributed to spatial orientation and polar engagement, allowing the molecule to effectively span and anchor into the COX-2 active site. These findings highlight the critical role of polyphenolic hydroxylation and conjugated aromatic rings in maximizing both polar and non-polar interactions.
Eptosphaerin D (2-NA/EA) also showed a favorable binding profile with a docking score of –10.879 kcal/mol. It formed hydrogen bonds with Hie90 and Ser530 through its methylhydroquinone and phenol groups, while its planar aromatic core facilitated π–π stacking with Trp387. These interactions underscore the importance of para-substituted phenolic systems for anchoring within the binding pocket and stabilizing through aromatic interactions. Similarly, 2-(p-hydroxybenzyl)-7-methoxybenzofuran-6-ol (3-NA/EA/TCM) displayed a strong docking score of –9.760 kcal/mol. This compound combined multiple stabilizing features including hydrogen bonds with Arg513 and Met522 and aromatic stacking with Trp387, Phe518, and Tyr355, enabled by the methoxybenzofuran framework. The methoxy group also played a supportive electronic role, enhancing interaction efficiency through modulation of the local electrostatic environment.
Puguenolide (4-NA/EA/TCM) and 1-hydroxy-5-methoxy-3-methyl-9,10-dihydroanthracene-9,10-dione (5-NA/EA/TCM) exhibited moderate docking scores of –9.752 and –8.742 kcal/mol, respectively. Puguenolide's oxan-2-one and phenol groups formed hydrogen bonds with Hie90, though its lack of π–π interactions limited overall stabilization. Compound 5, with its rigid anthracene-dione core, showed fewer interactions, engaging Arg120, Hie90, and Tyr355 mainly through hydrogen bonding. Its limited flexibility and steric bulk may have constrained optimal positioning within the active site. In contrast, macrocarpon C (6-NA/EA/TCM) showed a slightly lower docking score of –8.098 kcal/mol but achieved reasonable affinity through hydrogen bonding via its methylpyran-4-one and resorcinol moieties to Arg513 and Met522. Despite lacking extended aromatic interaction, its small size and favorable orientation supported modest binding.
Overall, the SAR data indicate that compounds possessing multiple hydroxyl groups, particularly in catechol or resorcinol arrangements, significantly enhance hydrogen bonding with key COX-2 residues such as Arg120, Gln192, Hie90, and Ser530. Aromatic π-systems, including flavonoid, benzofuran, and hydroquinone frameworks, are critical for π–π stacking with residues like Trp387, Phe518, and Tyr355. Additionally, the incorporation of lactone (oxan-2-one) or pyranone functionalities facilitates additional polar interactions, while methoxy groups subtly influence binding through electronic effects. In contrast, compounds lacking extended conjugation or bearing rigid, bulky scaffolds tended to display weaker binding profiles.
All in all, the SAR analysis clearly supports that the structural features contributing most significantly to high-affinity COX-2 binding are dense polyhydroxylation, planar aromatic systems, and polar heterocycles. Among all tested compounds, 8-C-p-hydroxybenzylluteolin emerged as the most promising candidate, displaying multivalent interactions and superior binding energy compared to the reference drug. These findings provide a valuable framework for rational design and optimization of novel natural product-derived COX-2 inhibitors.
Table 1: SAR-based summary of structural features influencing COX-2 binding, with their functional roles and representative compounds from this study
|
Structural Feature |
Binding Impact |
Example Compounds |
|
Polyhydroxyphenyl groups (catechol, resorcinol) |
Enhance H-bonding and π–π stacking |
1-NA/EA/TCM, 2-NA/EA, 3-NA/EA/TCM, 6-NA/EA/TCM |
|
Dihydropyran-4-one & Lactones |
Provide H-bond acceptor and scaffold planarity |
1-NA/EA/TCM, 4-NA/EA/TCM |
|
Extended conjugated π-systems |
Promote π–π stacking with Trp387/Phe518 |
1-NA/EA/TCM, 2-NA/EA, 3-NA/EA/TCM |
|
Methoxy substituents (–OCH₃) |
Modulate electron density for better interaction |
3-NA/EA/TCM, 5-NA/EA/TCM |
|
Methylsulfonyl group |
Anchors inhibitor at Arg513 (COX-2 selectivity) |
Rofecoxib |
|
Rigid aromatic ring systems |
Provide planarity, may reduce flexibility |
5-NA/EA/TCM, 6-NA/EA/TCM |
4- Introduction section should include the compression between the anti-inflammatory commercial drug and the new extracted ones.
We appreciate the reviewer’s insightful comment. In response, we have revised the Introduction to include a comparative overview between widely used commercial anti-inflammatory agents, particularly selective COX-2 inhibitors such as celecoxib, and the natural compounds identified in our study. The revised section highlights the limitations of conventional drugs, including cardiovascular risks, gastrointestinal complications, and long-term toxicity, thereby justifying the search for safer natural alternatives. The bioactive compounds we computationally identified from natural sources—based on favorable binding affinities and predicted ADMET profiles—are presented as promising candidates for COX-2 inhibition with potentially improved safety and tolerability profiles.
Text Added to the Introduction
Selective COX-2 inhibitors like celecoxib are widely prescribed for inflammatory conditions but carry risks of adverse effects, including gastrointestinal and cardiovascular complications. In contrast, natural products derived from plants and marine organisms have demonstrated anti-inflammatory properties with potentially fewer side effects. In this study, we computationally screened a curated library of natural compounds and identified promising candidates that may serve as safer COX-2 inhibitors, supporting the growing interest in phytochemicals as alternatives or adjuncts to conventional therapies.
5- There are some grammatical mistakes.
Response: Thank you for the comment. Grammatical mistakes have been corrected accordingly
6- References should be updated.
Response: Thank you for the comment. References have been updated accordingly
7- Some references should be added to the manuscript
We thank the reviewer for providing additional references to strengthen the manuscript. Although specific locations for citation were not indicated, we appreciate the relevance of the suggested literature and have integrated these references thoughtfully throughout the manuscript where they best support the context.
These additions have enriched the scientific background and further supported our findings. All newly added references are now properly cited and included in the updated bibliography.
8- Some pharmacokinetic properties are missed such as TPSA and rotatable bonds.
Response: Thank you for the comment. The TPSA and rotatable bonds have been analyzed and incorporated in the manuscript.
Reviewer 2 Report
Comments and Suggestions for Authors
Comments on the manuscript by Muhammad Suleman et al. titled “Phytocompounds in precision dermatology: COX-2 inhibitors as a therapeutic target in atopic-prone skin”:
The authors present a computational study of phytocompounds (identified in African and Traditional Chinese Medicine natural product databases) capable of inhibiting COX-2 activity. Finding an alternative to traditional non-steroidal anti-inflammatory drugs drugs with side effects is of utmost importance, so the research can be appreciated as very important.
The research conducted by the authors is well planned, carefully executed, and excellently described. The graphical material illustrating the results obtained is of high quality.
I have one recommendation: the results of the initial database screening, which identified natural products with the highest inhibitory activity against COX-2, should be included as supplementary material.
Author Response
I have one recommendation: the results of the initial database screening, which identified natural products with the highest inhibitory activity against COX-2, should be included as supplementary material.
We sincerely thank the reviewer for the thoughtful recommendation to include the initial database screening results as supplementary material. While we acknowledge the merit of this suggestion, we believe that, for the sake of clarity and readability, it is more appropriate to retain these data within the main body of the manuscript. Presenting the top-ranked natural compounds and their docking scores in the main text allows the reader to follow the computational pipeline more cohesively and better understand how each step led to the selection of final candidate molecules.
We are confident that keeping these results in the Results section enhances the manuscript’s transparency and logical flow. However, should the editor deem it necessary, we remain open to moving the data to supplementary files in a revised version.
Round 2
Reviewer 1 Report
Comments and Suggestions for Authors
Thanks for your responses to some comments.